# Active Transportation to School. Utopia or a Strategy for a Healthy Life in Adolescence

**DOI:** 10.3390/ijerph18094503

**Published:** 2021-04-23

**Authors:** Nuno Loureiro, Adilson Marques, Vânia Loureiro, Margarida Gaspar de Matos

**Affiliations:** 1Projeto Aventura Social, Faculdade de Motricidade Humana, Universidade de Lisboa, Estrada da Costa, 1499-002 Cruz Quebrada, Portugal; nloureiro@ipbeja.pt (N.L.); mmatos@fmh.ulisboa.pt (M.G.d.M.); 2Faculdade de Medicina/ISAMB Centro de Investigação Apoiado Pela Universidade de Lisboa, 1649-028 Lisboa, Portugal; amarques@fmh.ulisboa.pt; 3Instituto Politécnico de Beja, Departamento de Artes, Humanidades e Desporto, 7800-295 Beja, Portugal; 4CIPER, Faculdade de Motricidade Humana, Universidade de Lisboa, Estrada da Costa, 1499-002 Cruz Quebrada, Portugal

**Keywords:** adolescent, walking, health behaviors, health promotion, physical activity

## Abstract

The way adolescents travel to school can be an important contribution to achieving their daily physical activity recommendations. The main objective of this research is to know which variables are associated with the mode of transportation used to and from school by Portuguese adolescents. The 2018 Health Behaviour in School-aged Children questionnaire was applied to 5695 adolescents with an average age of 15.5 (SD ± 1.8), 53.9% of whom were female. The associations were studied by applying χ^2^ tests and multivariate logistic regression models. In this study, 36.5% of the participants reported walking or cycling to school. Active transportation to school is associated with age (OR = 1.3; *p* < 0.05), sufficient physical activity (OR = 1.2; *p* < 0.05), adequate number of hours of sleep (OR = 1.2; *p* < 0.05), perception of happiness (OR = 1.2; *p* < 0.05) and living near the school (OR = 2.4; *p* < 0.05). The results revealed that adolescents’ choice to travel to/from school using an active mode of transportation increased with age, physical activity, hours of sleep, perception of happiness, and living near the school.

## 1. Introduction

The positive relationship between physical activity and health is very well documented, and there is a recommendation of 60 min or more of daily physical activity for adolescents [1]. However, only 15.6% of Portuguese adolescents meet the recommendations [2]. In order to reverse this indicator, several authors [3,4,5] showed that active transportation (AT) could be an important strategy to increase daily physical activity. In addition to this, AT has numerous health benefits [6], such as those related to a lower percentage of body fat [7,8] and higher cardiorespiratory fitness [7,9]. It is also associated with a better cardiometabolic health profile [10]. There are also other aspects that are increasingly important in modern society, such as economic and social issues and the ecological co-benefits [11,12]. From an educational perspective, AT stimulated at an early age [13] can help consolidate this behavior throughout life [14].

Older adolescents have greater autonomy [15], are less likely to be conditioned by their parents’ opinions about the way they move, and present greater practice of AT [16]. Considering gender, there are contradictory results because some studies do not show a significant difference between boys and girls [17], while others report that boys were more likely to engage in AT to school than girls [18].

There are other interesting relationships related to the high frequency of AT, such as a greater practice of physical activity [19], higher self-efficacy, social support from peers, and enjoyment of physical activity [20].

Sleep time seems to be a structural factor for adolescents’ health [21]. According to the recommendations, the adequate range of sleep duration for children is from 9 to 11 h, and for young and old adolescents, it is from 8 to 10 h [22]. Adolescents who sleep the recommended number of hours are more likely to do more AT to school [23].

The way adolescents perceive their residential context has been one of the most important factors in promoting physical activity [24], and it is particularly relevant regarding the travel mode choices to/from school [18] and, in particular, security issues [25].

The characteristics of the school setting [26] and the distance that adolescents have to travel from their homes are two of the most consistent indicators for adopting this behavior [27]. Some authors consider distance as the clearest [27] and biggest predictor associated with AT in adolescents [28,29].

Given the possibility that AT to school could be an effective approach for the promotion of adolescent health, this research intends to find out which variables are associated with the used modes of transportation to and from school by Portuguese adolescents.

## 2. Materials and Methods

### 2.1. Participants and Procedures

This study is based on data from the cross-sectional Health Behavior in School-aged Children (HBSC) Portugal survey, conducted in collaboration with the World Health Organization (WHO) Regional Office for Europe in 2018 [30]. HBSC is an international epidemiological school-based survey carried out every 4 years in 44 countries and regions across Europe and North America and surveys adolescents’ health and health behaviors.

The participants were recruited via a clustered sampling design (the sampling unit was the class) to meet the required number of students for each school grade from selected national public schools, according to standard guidelines from the HBSC/WHO survey protocol [31]. All schools gave their approval, legal guardians gave signed informed consent, and students provided assent.

Web-based online surveys were administered during class time by trained teachers. Participation was voluntary, and questionnaires were answered anonymously. The questionnaire application took approximately 60 min. The research was in accordance with the Ethical Committee of Porto Medical School and the National Data Protection System and had the approval of the Ministry of Education.

In order to obtain a representative sample of the Portuguese school population, 42 school groups from all over the continental country (5 school regions) were selected, a total of 387 classes. The sample is representative of the years of schooling in the study. This study comprised 5695 Portuguese adolescents (53.9% females, *M*_age_ = 15.5 ± 1.8 years), randomly selected at the national level and included in the HBSC study in 2018. Adolescents were in the eighth grade (n = 2766, M_age_ = 14 ± 0.83 years), tenth (n = 1711, M_age_ = 16.1 ± 0.9 years) and twelfth year of schooling (n = 1218, M_age_ = 18.0 ± 0.8 years).

### 2.2. Measures

#### 2.2.1. Outcome—Active Transportation to School

The outcome of interest was regular engagement in AT to schools, such as walking and cycling. Participants answered the following HBSC survey question: “How do you go from home to school every day (and from school to home)?” with the following options: (a) public transportation; (b) car; (c) motorcycle; (d) walking; (e) bicycle; (f) other. Responses were grouped dichotomously: those who answered “walking” or “bicycle” were categorized as students who regularly engage in AT to school, while those who answered public transportation, car, and motorcycle were categorized as individuals who do not engage in AT to school. Participants who answered “other” to this question were excluded to minimize possible misclassification.

#### 2.2.2. Individual Correlates

Participants reported their age (year and month of birth), gender (boy/girl), and schooling. The level of physical activity of adolescents was assessed by asking students: “Outside of school hours, in your free time, how many hours in a week do you usually exercise so much that you get out of breath or sweat?” (responses: “none”, “half an hour”, “about 1 h”, “about 2 to 3 h”, “about 4 to 6 h”, “about 7 h or more”). Participants reporting 4 or more hours per week were classified as achieving sufficient MVPA. The amount of sleep during the week was reported as follows: “On average, how many hours do you sleep a night?” (responses: “5 h or less”, “6 h”, “7 h”, “8 h”, “9 h”, “10 h or more”). The students were categorized as having “adequate sleep duration” when they met The Sleep Foundation recommendations for health and “non-adequate sleep duration” when they did not meet these recommendations. According to the recommendations, the adequate range of sleep duration for children is from 9 to 11 h, and for young and old adolescents, it is from 8 to 10 h [22].

#### 2.2.3. Perceptions Correlates

The data on how adolescents perceived their health were obtained through the question: “Would you say that your health is…?” (responses: “Excellent”, “Good”, “Reasonable”, “Bad”). The students were categorized in “Good” to “Excellent” perception of their health and in “Reasonable” to “Bad” perception of their health. The perception of the neighborhood where they live was obtained by the question: “We would like to ask you some questions about where you live. Choose the option that best describes your opinion: In general, I feel safe in my area” (responses: “Yes, it’s very good”; “Yes, it’s good”; “It’s reasonable”; “Not much good”, “It’s nothing good”). Adolescents reporting “Not much good” or worst were classified as do not feel safe. The question about their happiness perception was: “How do you feel about life in general?” (responses: “I feel very happy”, “I feel happy”, “I feel a little happy”, “I feel unhappy”) and was recoded into a binary variable (0 = unhappy vs. 1 = happy).

#### 2.2.4. School Correlates

The adolescents were able to give their opinion about their school with the following question: “What do you feel about the school?” (responses: “I like it a lot”, “I like it more or less”, “I do not like it very much”, “I do not like it at all”) and this was recoded into a binary variable (0 = no vs. 1 = yes). The proximity of the student’s home to the school was evaluated in the question: “How much time do you take from your home to your school? Consider the house where you live most of the time”. According to the McDonald [32] recommendation, the variable was dichotomized in “Near” (when the time was reported as less or equal to 16 min) or “Faraway” (longer than 16 min).

### 2.3. Statistical Analyses

Descriptive analyses for all variables (mean, standard deviations, and percentages) were calculated for the total sample. The chi-squared test was used to investigate the statistically significant differences between each mode of commuting to the school, the individual characteristics (age, schooling, gender, physical activity, and sleep duration), perceptions (health, safety of the neighborhood, and happiness) and the school-related characteristics (like the school, school proximity), for the total sample. Binary multilevel logistic regression analyses were performed to assess the associations between each mode of commuting and the gender, age, schooling, physical activity, sleep time, happiness, and proximity of the school. Results are presented as numbers, proportions, and/or odds ratios (OR) and their 95% confidence intervals (95% CI). All data analyses were completed using the Statistical Package for Social Sciences (SPSS, version 27.0; IBM SPSS Inc., Chicago, IL, USA) and α was set at 0.05. 

## 3. Results

The descriptive characteristics of the study sample are shown in Table 1.

We verified that 53.9% of the subjects inquired were girls (46.1%, boys), and the average age was 15.5 years. We found that 37.6% of the young people fulfill the recommendations of the practice of daily physical activity, 36.5% reported choosing AT when traveling to school, and 50.1% slept a sufficient number of hours for their age. Regarding perceptions, the vast majority of adolescents reported having good health (85.5%), considered themselves happy (85.4%), and 61.2% revealed feeling safe in their neighborhood. Most teens said they liked the school (64.5%) and took 15.3 min (average) on their usual home-to-school route.

The results of the chi-square test (χ^2^) shown in Table 2 determined that there were no significant differences regarding AT according to gender, perception of health, perception of neighborhood security, and the opinion about the school.

Significant differences were identified among the age groups, and the older adolescents were (15.2%) reported to do more AT. The adolescents who did not comply with the daily physical activity recommendations most refrained from performing AT (22.1%). It was the adolescents who slept the appropriate number of hours per night (18.6%), felt happy (28.2%), and lived near the school (30.5%) that carried out more AT.

Table 3 presents the adjusted odds ratio values of the statistical multivariate logistic regression and the variables associated with AT. The oldest adolescents were more likely to choose AT (OR = 1.3; *p* < 0.05). There was evidence that healthy behaviors, such as sufficient physical activity (OR = 1.2; *p* < 0.05) and the appropriate sleep time (OR = 1.2; *p* < 0.05), were positively associated with the probability of the adolescents adopting AT. Feeling happy (OR = 1.2; *p* < 0.05) and living near the school (OR = 2.4; *p* < 0.05) also contributed to increasing the probability of adopting AT.

## 4. Discussion

This study examined the variables that may be related to the practice of AT to school among Portuguese adolescents. The results revealed that AT increases in adolescents with older age that practice enough physical activity, sleep the appropriate number of hours, reveal a happiness perception and live near the school.

Considering that Portuguese adolescents, especially girls, present one of the lowest levels of physical activity practice in the countries participating in the HBSC study [33], it is relevant to understand the variables that contribute to this type of mobility and then implement the best strategies that contribute to achieving the goals defined in the Portuguese Strategy for the Promotion of Physical Activity, Health and Well-Being [34]. On the other hand, AT can improve physical activity levels [35,36], cardiorespiratory fitness levels, and body composition measures [9] while suppressing sedentary behavior. Promoting AT not only suppresses or reduces sedentary behavior but replaces it with a physical activity behavior.

The presented study reveals that walking or cycling to school has a low adherence by adolescents (36.5%), but this is similar to the values obtained (35.8%) by a study carried out in the same geographic context [37]. In a study conducted with young people from a very distant country such as New Zealand, the percentage obtained was 37% [38], but this is far from the 60% reported in Wang and colleagues’ [39] research.

As is referred to in another study, this active behavior is reported more often by older adolescents. According to Panter et al. [40]., age is an important moderator for adolescent AT since the older ones may have fewer personal safety concerns due to greater autonomy and less dependence on parents. On the other hand, they have less parental control, which makes a significant contribution to the decisions young people make on the way they move [41].

The number of hours that young people sleep is associated with an increased likelihood of carrying out AT and is in line with the results of other researchers [23]. Van den Bulck [42] mentions that the morning fatigue caused by a lack of sleep could mediate the association between sleep duration and AT, choosing inactive modes of commuting to school rather than active modes.

Young people who are more active are more likely to perform AT, which is confirmed in other studies [20]. However, many young people may forget to include this behavior when assessing their amount of daily physical activity.

Happiness is an example of a positive construct of mental health that may be promoted by physical activity. Happiness could also increase resilience to emotional perturbations, and it was found that only walking and vigorous and intense physical activity had small associations with happiness [43]. This situation seems to be evident in the data obtained in this investigation.

We verified that the greatest predictor of AT was living near the school, which establishes a clear association between distance and mode of commuting used [27]. Parents perceive the barrier of the distance to school being too far/travel taking too much time as significantly influential in decreasing children’s use of AT on the journey to school [25]. Interventions to promote AT should first target children living within a 900-m distance (0.56-mile distance) from school and adolescents living within 1300 m (0.81 miles) [27].

The present study has some aspects that we consider relevant and should be mentioned. First, the size and representativeness of the sample should be valued since the results of inferential statistics can be extrapolated to the national context where the study was performed and of which it is representative, with a reduced error. One of the limitations of this study is related to the fact that the questions used did not address the characterization of AT, which may have caused interpretation problems for the subjects under inquiry. Another restraint involved the variables related to perceptions, always conditioned by a high degree of subjectivity, for which the complementary use of instruments that assess AT directly is recommended.

## 5. Conclusions

This work allowed for a deeper exploration of the variables associated with AT reported by Portuguese adolescents in the 2018 HBSC study. The results show a 36.5% prevalence of AT, and we conclude that the choice of making the journey to/from school in an active mode increases with older age, enough physical activity, appropriate hours of sleep, happiness perception, and living near the school. We found no association between AT and gender. AT could be adopted as a strategy in which adolescents can incorporate physical activity into their daily routines in order to achieve the recommended levels and, subsequently, possibly increase adolescents’ health. We also found that living closer to the school is a facilitator in adopting AT. Educational authorities should make efforts to implement actions that encourage the adoption of this behavior.

## Figures and Tables

**Table 1 ijerph-18-04503-t001:** Descriptive characteristics for the participants and the variables in this study—HBSC Portugal (N = 5695).

Items Studied	n ^1^	Mean (SD)	%
**Individual**			
Age (years)	5695	15.5 (±1.8)	
Gender	5695		
Female	3067		53.9
Male	2628		46.1
Grade	5695		
8th year	2766		48.6
10th year	1711		30.0
12th year	1218		21.4
Physical Activity	5564		
Insufficient (<4 days/week)	3470		62.4
Enough (≥4 days/week)	2094		37.6
Sleep time	5077		
Inappropriate (<8 h)	2224		43.8
Appropriate (≥8 h)	2853		56.2
Active Transportation	4276		
No	2716		63.5
Yes	1560		36.5
**Perceptions**			
Perception of health	5638		
Reasonable to bad	817		14.5
Good or excellent	4821		85.5
Perception of neighborhood’s security	4273		
Bad	1660		38.8
Good	2613		61.2
Perception of happiness	5695		
Happy	4290		85.4
Unhappy	1405		14.6
**School Level**			
Likes school	5524		
Yes	3565		64.5
No	1959		35.5
Proximity to school (min.)	4347	15.3 (±13.9)	

^1^ Number of adolescents that completed the survey item.

**Table 2 ijerph-18-04503-t002:** χ^2^ Inferential analysis of the adolescents’ gender, grade, age, physical activity, sleep time, perception of health, of neighborhood’s security, of happiness, likes school and proximity to school according to active transportation (n = 4276).

Variables	Active Transportation			
	No		Yes	
	n	%	n	%
**Gender**				
Female	1534	35.9	833	19.5
Male	1182	27.6	727	17.0
**Grade ***				
8th year	1251	**29.3**	635	**14.9**
10th year	865	20.2	510	11.9
12th year	600	**14.0**	415	**9.7**
**Age ***				
13–14 years	1122	**26.3**	535	**12.6**
15–16 years	676	15.8	370	8.7
>16 years	913	**21.4**	652	**15.2**
**Physical Activity ***				
Insufficient	1741	**40.7**	944	**22.1**
Enough	975	**22.8**	616	**14.4**
**Sleep Time ***				
Inappropriate	1147	**26.9**	761	**17.8**
Appropriate	1564	**36.7**	795	**18.6**
**Perception of Health**				
reasonable to bad	379	8.9	236	5.5
good or excellent	2337	54.7	1324	31.0
**Perception of Neighborhood’s Security**				
Bad	971	24.2	599	14.9
Good	1574	39.2	868	21.6
**Perception of Happiness ***				
Happy	2206	**51.6**	1206	**28.2**
Unhappy	510	**11.9**	354	**8.3**
**Likes School**				
Yes	1797	42.0	1011	23.6
No	919	21.5	549	12.8
**Proximity to School ***				
Distant	837	**19.8**	264	**6.2**
Near	1845	**43.6**	1290	**30.5**

***** χ^2^ significant values for *p* < 0.05; Adjusted residuals ≥|1.9| are considered significant (in bold).

**Table 3 ijerph-18-04503-t003:** Explanatory logistic regression of the Portuguese adolescents’ modes of AT to school (n = 4276).

Variable	Active Transportation OR (95% CI)
**Gender**	
Male	0.9 (0.8–1.1)
**Age ***	1.3 (1.2–1.4) *
**Physical Activity ***	
Enough	1.2 (1–1.3) *
**Sleep Time ***	
Appropriate	1.2 (1.1–1.4) *
**Perception of Happiness ***	
Happy	1.2 (1.1–1.5) *
**Proximity to School ***	
Near	2.4 (2.0–2.8) *
Constant	0.006

Note: * Girls were the reference group; the insufficient practice of physical activity is the reference group; inappropriate sleep time was the reference group; the perception of unhappiness was the reference group; living far from school was the reference group. Values in bold mean significant results *p* < 0.05; CI indicates confidence intervals; OR means odds ratio.

## Data Availability

The data presented in this study are available at http://aventurasocial.com/ (accessed on 10 March 2021).

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
