# Peer review of "Active Transportation to School. Utopia or a Strategy for a Healthy Life in Adolescence"

_ijerph, 2021, doi:10.3390/ijerph18094503_

Round 1

Reviewer 1 Report

The study described in the article is interesting from a scientific and cognitive point of view. A large research sample means that the study deserves an in-depth analysis. From my point of view, there are several aspects that require clarification and perhaps modification of the text. In my opinion, the research procedure is insufficiently explained. The relatively enigmatic subsection 2.1 requires elaboration, perhaps proposing a diagram explaining the research procedure. Second, it is absolutely necessary to elaborate on the conclusions. It was written casually, no reference was made to other achievements in this subject, and no description was made of the extent to which the research carried out extends scientific knowledge. 

Author Response

We would like to thank the time and effort that the reviewer dedicated to providing feedback on our manuscript and valuable improvements to our paper.

Reviewer 2 Report

Dear authors,

I find the subject of your study very interesting, as starting the morning with physical activity improves compliance with the physical activity recommendations for this age group. In addition, the changes that society is making are not encouraging, use of electric scooters, ...

After I have read carefully your manuscript, I have noticed some points that I think you should revise.

In the type of transport, you have not introduced the use of conventional scooters, could it be considered active transport? is its use in Portugal minimal?

The section Results should be written in the past tense. In the manuscript it appears in the present tense.

In table 2 you should always use a decimal to keep the format, for example: Male % 17, you must write 17.0. Check the whole table.

In table 1 there are 3 age ranges, but in table 3 there are none. Is it the same OR for the 2 groups (except the reference group)? It would be interesting if both groups appear, to know if the OR is different. Why didn't you use the grade variable in the multivariate regression model?

The discussion should not start with the objective, but with the most important results. It would be interesting to have a wider discussion, and with some concepts related to physical activity and health improvement (obesity, sedentary lifestyle, ...). Also, it would be interesting to name the measures that are being used in Portugal to correct sedentary lifestyle, especially if there is any measure on active transport, and if any of the studied variables could influence it.

Thank you very much for the opportunity to review your manuscript.

Kind regards.

Author Response

We would like to thank the time and effort that you dedicated to providing feedback on our manuscript and valuable improvements to our paper.

Reviewer 3 Report

Dear Editor,

Dear Authors,

Thank you for the opportunity to prepare my opinion about an article titled: “Active transportation to school. Utopia or a strategy for a healthy life in adolescence”. 

Authors took into consideration actual and very important elements of life – physical activity. PA is one of the most important components of healthy life and needs to be systematically developed and kept on an appropriate level from early age to the end of life.
The conclusions are very important points for wider this information and applying it into schools and systems of education. It is an important way especially in a situation when most people on every level of age use too much contemporary, latest technological developments which could be dangerous for health in the long perspective of independent life.

General comments:

The aim of the study is clear and right presented. and using data collected by good recognized tools like HBSC questionnaire let to state that the basement of the article is appropriately designed. 

However, on this background it is possible to find numerous confusing elements which I will present below.

The first general comment is that readers lose orientation who are the Subject of investigation Female or Male or Both. In the abstract we are able to meet information about Females (line 18) and in the next part of the article we suspect that general analysis is presented only for Males. It must be improved and clearly explained.

I am not native English speaker but in some places I would suggest more detailed language advice.

Detailed comments:

Introduction

In my opinion is well designed and describes base information to justify the eim of the study presented in the last paragraph. (line 59-61) 

Materials and Methods

2.1. Participants and procedures
Presented in the first part information (lines 64-76) describe tools and general idea of data collection. No comments.
However, lines 77 to 83 bring limited information about Subjects. Is the sample of 5695 the only female or also male? Authors probably forgot to present information about boys which have to be added to give a chance to have a clear picture of all participants without guesses.
All information should be present for both sexes: number of male and female, age, etc.
Lines 80 - 83 need to be changed comma for dot in the decimal numbers.
Line 82 - (miracle multiple?) Authors presented "n=17111" and probably it should be 1711.    

Other parts are presented generally right and clear.

2.3 Statistical analysis
Lines 132 - 133 "Descriptive analyses for all variables (mean, standard deviations, and percentages) were calculated for the total sample" - my suggestion is to present separately for Females and Males. It will bring more interesting information and prepare better proposition for representative of different sex group because as were stated in numerous researches - boys and girls are different in many variables and we need to take it into consideration during analysis and during programs design to stimulate proper PA..

Is not clear how you prepared analysis for both Females and Males or separately, especially that in the tables you presented data in one way and another time differently. 

 Please set and describe one way of analysis and keep it in all analysis and in the tables. 

Results
Line 145 and next - Table 1.

Please prepare a better version to describe data presented in the table. Title should be clear independently from the text and way of presentation. What did you present? For whom? Total sample or only for Male... ?
In the Abstract and in the sub-point 2.1 Participants we were able to read the only about Females and here suddenly the only Male!
Is data presented in table 1 dedicated only for Boys? What was SD for the age? The age was calculated for all 5695 groups like you stated in the Statistical analysis (lines 132-133) or in the table 1 the only for boys? Because in the lines 80-83 we have probably analyzed Girls ?! Writing these sentences I have problems keeping concentration and I am confusing who is described and where?
What about Females? In the abstract you presented just % of Female and here Male? Where did you present data about Females? the only in the lines 80-83? I did not find it in any other tables!?

My suggestion is to prepare one table and present this same data in separate columns for boys and for girls and in third one altogether. And then when you check statistical significance or lack of it you will be able to decide how presented results. Current way of data presentation in the tables is not clear enough and description of it in the Methods is confusing and needs to be improved.  It must be consistent especially for others.  

Line 147 and in table 1 "N1" - "Number completing the survey item." - is it concerned Male or Female or both - what was the difference? Are there a number of collected questionnaires? - it should be better described in the title of the table.
Please remember about it also in the next tables.

Line 149 - "We found that only 37.6% of the young people..." - only or till? please do not judge the numbers; it could depend on context and interpretation. Researchers should be independent. Next: Boys or Girls or both? It must be clear.

line 164 Table 2
Please change title for more clear.
At last Authors presented both sexes in one table together! Keep this way! However, I am not sure whether analysis presented concerning Boys or Girls or Both?
How did you calculate significance e.g. for Grade - between "Yes" and "No"? or among grades or both or also taking into consideration sexes? Even the description below table 2 did not help to understand it.

Line 165 - "Table 3 shows the adjusted..." - The table has not so much power to show anything. Please try in a different way, e.g.: In the table 3 were presented results of adjusted...
Line 166 - "showing..." try to discuss with Native speakers to improve text. I am not also native English speaker but always try to discuss and use professional support for better presentation of my/our thoughts.

Line 172 - Table 3
Please improve title for more clear.
As we can see we returned to Male. What do you mean about "Girls are the reference group"  - in my opinion norms or recommendations should be the reference's group.

My suggestion is to present both sex as separate results in relation to references value e.g. WHO or other standards. 

Discussion

Lines 174 - "The purpose of this study was to find out the variables that may be related to the practice of AT to school among Portuguese adolescents" my suggestion to add: who were observed in 2018 during HBSC research.
However, Authors should be aware that in the current version it is only half of true and I have no clear idea about whom boys or girls or both and who were excluded or included. Why? Because there are not clear and established descriptions from Abstract through Methods through Results to Discussion.
And Discussion should be improved according to the new version of data analysis and data presentation in the Results. Please let me, in this situation, not prepare wider comments for this version. My suggestion is, however, to concentrate a little more on relation "my results and references" which improve the quality of this part.

Line 220 and next - Conclusions
It is a general interesting summary which would be much more interesting when Authors would present differences among Females and Males and try to find common variables and significant differences. Please try to do it in the next version of your manuscript.

References.

Please once more carefully check some letter mistakes, omitted numbers of pages, etc.  

In conclusion. Article has interesting potential but current versions contain too many confusing elements to be accepted for publication which should be improved.
Please meet my comments and suggestions as friendly and supporting in not easy scientific writing art. Good luck and I am waiting for a new version of your interesting idea.

Author Response

(The authors gave the same response as above.)

Round 2

Reviewer 1 Report

Dear Authors,

thank you for making changes to the article suggested in the reviews. Currently, the article is of a much better level than originally, and I believe that it is suitable for publication.

The introduction and conclusions could be improved based on the best publications on the subject, but I am satisfied with the current level.

Good luck!